# Gauge Symmetries for Efficient Zero-Knowledge Proofs of Transformers

## Abstract

We introduce **GaugeZKP**, a symmetry-aware verification framework for Transformers that exploits the maximal gauge group of attention. For canonical models the maximal group is $G_{\max} = ((\mathrm{GL}(d_k))^h \times (\mathrm{GL}(d_v))^h) \rtimes S_h$; with RoPE (LLaMA/Qwen), the Q/K action reduces to the rotary commutant $C_{\mathrm{RoPE}}$. We operationalize this via a one-time proof of gauge equivalence (PoGE) to a canonical model and per-inference proofs (PoVI) on that canonical model. On Halo2 circuits, canonicalization reduces *model-level prover gates/constraints* by up to $\approx 26\%$ without changing model function; because this optimization is upstream of the prover, pairing with frameworks like EZKL/zkVM further reduces proving *time/memory* on the smaller circuit. Analytically, the savings *multiply* with parameter tying in grouped/single-query attention (GQA/MQA) and with MoE sparsity, since PoGE/PoVI scale with the number of *distinct* parameter blocks rather than with the head count.

## 1 Introduction

Deploying large language models in sensitive settings creates a need for *verifiable inference*: proofs that outputs were computed correctly without revealing proprietary weights or private inputs. Zero-knowledge ML (ZKML) offers this, yet current systems struggle to scale to Transformer architectures.

Existing frameworks compile networks into polynomial constraint systems; prover cost is dominated by the number of constraints (roughly linear in parameter count). Cryptographic advances (e.g., lookups, sumcheck, commitments) accelerate the *protocol* layer, but they do not remove the *model-level* redundancy intrinsic to attention—leaving many constraints structurally unnecessary.

**Key idea (GaugeZKP).** Exploit attention's *gauge symmetries*. Many parameterizations implement the same function. We rewrite deployed weights into a *canonical* form (constructed per head; see §3.4) without changing the model. A one-time *Proof of Gauge Equivalence* (PoGE) binds deployed and canonical weights; thereafter, per-inference *Proofs of Verifiable Inference* (PoVI) run only on the canonical model. Because this optimization is upstream of the prover, it composes with protocol-level speedups.

**Scope and guarantees.** We certify *exact* functional equivalence (no approximation); privacy follows from the zk proof system. Attention remains quadratic in sequence length. Canonicalization relies on full-column-rank projections and numerically stable QR/SPD roots; fixed-point precision uses scale $2^{16}$ (deterministic choices yield identical outputs across implementations).

**Empirical summary.** On Halo2 circuits, canonicalization reduces *model-level prover gates/constraints* by up to $\approx 26\%$ without changing model function. Since circuits are smaller, pairing with EZKL/zkVM further reduces *time/memory* on the same instances (e.g., GPT-2 Base). Results use identical models and inputs; we report circuit-level units in the body and appendix.

**Contributions.** (i) *Theory.* Complete, operational characterization of the maximal attention gauge group $G_{\max} = ((\mathrm{GL}(d_k))^h \times (\mathrm{GL}(d_v))^h) \rtimes S_h$; with RoPE the Q/K action restricts

to the rotary commutant $C_{\mathrm{RoPE}}$ (§3.3); depth-wise factorization $G_{\mathrm{model}} = \prod_{\ell=1}^{L} G_\ell$ (§3.1). (ii) *Canonicalization.* A per-head construction (VO thin-QR with positive diagonals; Q/K balancing; RoPE plane-wise scaling; deterministic head ordering) that preserves the forward map and is certifiable by PoGE (§3.4). (iii) *System.* A modular PoGE (one-time) + PoVI (per sequence) architecture binding deployed weights to the canonical model. (iv) *Empirics.* Up to $\approx 26\%$ fewer model-level gates on Halo2; when paired with EZKL/zkVM this translates into faster proving on the reduced circuit family. (v) *Analytics.* Savings *multiply* with grouped/single-query attention (GQA/MQA) and MoE sparsity, since PoGE/PoVI scale with the number of *distinct* blocks $(n_Q, n_K, n_V, n_O)$ rather than head count $h$ (see §A.3.1).

**Takeaway.** Gauge symmetry is a model-level optimization primitive for ZKML: it shrinks circuits *before* proving begins. RoPE primarily affects Q/K (modest per-head savings), while VO-side checks and structured $W_O$ target dominant costs; together with head tying, these effects yield larger model-level reductions without altering model behavior.

## 2 Background

**Transformer Parameterization.** Transformers process inputs through multi-head attention Vaswani et al. (2017), with per-head projections $W_Q^{(i)}, W_K^{(i)}, W_V^{(i)} \in \mathbb{R}^{d_{\mathrm{model}} \times d_k}$ and output matrices $W_O^{(i)}$. Attention scores are computed as $Q_i K_i^\top$, normalized via softmax, and applied to values before output projection. Modern models such as LLaMA employ rotary position embeddings (RoPE), which apply position-dependent rotations to queries and keys and constrain allowable parameter transformations.

**Gauge Symmetries in Transformers.** Many distinct parameterizations of a transformer block compute the same function. Unlike fully connected networks where symmetries are largely discrete unit permutations Grosse et al. (2012), transformers exhibit continuous symmetries within each head. Recent work has identified head permutation symmetries Malek et al. (2023) and explored completeness of invariant representations Kakarala (1992). The maximal gauge group can be characterized as

$$G_{\max} = \left( (\mathrm{GL}(d_k))^h \times (\mathrm{GL}(d_v))^h \right) \rtimes S_h, \tag{1}$$

acting by $(W_Q^{(i)}, W_K^{(i)}) \mapsto (W_Q^{(i)} A_i, W_K^{(i)} (A_i^{-1})^\top)$ and $(W_V^{(i)}, W_O^{(i)}) \mapsto (W_V^{(i)} C_i, C_i^{-1} W_O^{(i)})$, with $S_h$ permuting heads. For RoPE models, admissible $A_i$ must commute with rotation blocks, restricting the symmetry to $((C_{\mathrm{RoPE}})^h \times (\mathrm{GL}(d_v))^h) \rtimes S_h$. These symmetries imply large families of redundant parameters that do not alter model function.

**Zero-Knowledge Proofs for Machine Learning.** Zero-knowledge systems enable verifying inference correctness without revealing weights or inputs Goldwasser et al. (1985). Two dominant approaches exist: circuit-based systems, which compile networks into arithmetic constraints using frameworks like EZKL Henzinger et al. (2023), and zero-knowledge virtual machines, which prove execution of compiled inference programs using systems like RISC Zero's zkVM Team (2024a) and SP1 Labs (2024). Recent industrial deployments include zkLLM Network (2024) and Giza Tech (2024), demonstrating feasibility at moderate scale, though cost grows linearly with raw parameter count.

**Current Limitation.** Existing ZKML methods treat every weight as an independent variable. For a 110 million parameter transformer, this yields circuits of proportional size even though over one million degrees of freedom are functionally redundant under gauge symmetry. Thus, state-of-the-art proofs pay a large cost for verifying transformations that do not affect the model's output.

## 3 Gauge Group of Transformer Parameters

### 3.1 Formal Characterization

**Definition 3.1** (Head-wise gauge action)**.** *For each head $i$ with key/query width $d_k$ and value width $d_v$, the following invertible reparameterization preserves the attention block's input–output map (for all inputs, in exact arithmetic):*

$$(W_Q^{(i)}, W_K^{(i)}) \mapsto (W_Q^{(i)} A_i, W_K^{(i)} A_i^{-\top}), \qquad A_i \in \mathrm{GL}(d_k),$$

$$(W_V^{(i)}, W_O^{(i)}) \;\mapsto\; (W_V^{(i)}C_i, \; C_i^{-1}W_O^{(i)}), \qquad C_i \in \mathrm{GL}(d_v),$$

*optionally composed with a head permutation $\sigma \in S_h$ that reindexes the $(Q, K, V, O)$ blocks. With Rotary Positional Embeddings (RoPE), $A_i$ must commute with the plane-wise rotations (see Sec. 3.3); equivalently, on each $2\times2$ RoPE plane $p$,*

$$A_i^{(p)} = a_p I + b_p J, \qquad J = \begin{pmatrix} 0 & -1 \\ 1 & 0 \end{pmatrix}.$$

*This action preserves all logits $q_t^{(i)} k_s^{(i)\top}$ and the value–output map $\sum_s \alpha_{t,s}^{(i)} v_s^{(i)} W_O^{(i)}$, hence the layer output; it is compatible with residual connections and LayerNorm.*

**Proposition 3.2** (Single-layer maximality on the generic stratum). *Under generic conditions (full column rank for $W_Q^{(i)}, W_K^{(i)}, W_V^{(i)}$ and non-degenerate attention), the set of parameter transforms that preserve a single attention layer equals*

$$G_{\max} \;=\; \Big( (\mathrm{GL}(d_k))^h \times (\mathrm{GL}(d_v))^h \Big) \rtimes S_h, \tag{2}$$

*acting head-wise as in the definition above. With RoPE, the query/key factor $(\mathrm{GL}(d_k))^h$ is restricted to $(C_{\mathrm{RoPE}})^h$. For complete proof, see §A.1.1.*

**Corollary 3.3** (Depth-wise direct product). *In a residual Transformer with (pre-/post-)LayerNorm, the per-layer gauge actions preserve each layer's pre/post-LN interfaces and residual connections. With no cross-layer weight sharing, the model-level gauge group factorizes as a depth-wise direct product:*

$$G_{model} \;\cong\; \prod_{\ell=1}^{L} G_\ell. \tag{3}$$

### 3.2 Quantitative Redundancy

Per layer (attention submodule), the redundant degrees of freedom under the maximal gauge $G_{\max}$ are $h(d_k^2 + d_v^2)$; with RoPE, the Q/K part shrinks to $O(h\,d_k)$, yielding $h(d_k + d_v^2)$. In GQA/MQA, replace $h$ on the K/V terms by the group counts $(n_K, n_V)$.

See Table 3 in the appendix for *full-model* redundancy across common architectures; per-layer counts are obtained by dividing by $L$ (Cor. 3.3). FFN/MoE contributions are added separately, and if any projections are tied across depth, multiply only the untied parts.

### 3.3 RoPE Commutant Analysis

RoPE applies per–$2\times2$ rotations on the query/key features. Let

$$R_\theta \;=\; \begin{pmatrix} \cos\theta & -\sin\theta \\ \sin\theta & \cos\theta \end{pmatrix}, \qquad J \;=\; \begin{pmatrix} 0 & -1 \\ 1 & 0 \end{pmatrix}.$$

On each RoPE plane $p$, admissible gauge transforms $A_i^{(p)}$ must *commute* with *all* position rotations on that plane.

**Lemma (plane-wise commutant).** The real commutant of $\{R_\theta : \theta \in \mathbb{R}\}$ in $\mathbb{R}^{2\times2}$ is

$$\mathrm{Comm}(\{R_\theta\}) \;=\; \{\, aI + bJ : a, b \in \mathbb{R} \,\} \;=\; \{\, s\,R_\phi : s > 0, \; \phi \in \mathbb{R} \,\}.$$

*Proof idea.* From $X R_\theta = R_\theta X$ for all $\theta$, differentiate at $\theta = 0$ to get $XJ = JX$, whose real solution set is exactly $\{aI + bJ\}$.

**Block structure across $d_k$.** Decompose $d_k$ into $d_k/2$ RoPE planes. With a non-degenerate frequency schedule (distinct plane frequencies), the commutant is block-diagonal with independent plane-wise blocks $A_i^{(p)} \in \{a_p I + b_p J\}$. If some frequencies coincide, the commutant enlarges on the corresponding isotypic component (mixing is allowed within equal-frequency groups); our analysis and circuits assume the *generic* non-degenerate case used in practice.

**Dimension and parameterization.** Each plane contributes two real parameters $(a_p, b_p)$, i.e., $d_k$ real dof per head. Equivalently, $A_i^{(p)} = s_p R_{\phi_p}$ with scale $s_p > 0$ and rotation $\phi_p \in \mathbb{R}$.

**Corollary 3.4** (RoPE Gauge Group Reduction). *Under a non-degenerate RoPE schedule, the single-layer gauge group reduces to*

$$G_{RoPE} \;=\; \left((C_{RoPE})^h \times (\mathrm{GL}(d_v))^h\right) \rtimes S_h, \qquad C_{RoPE} \;\cong\; \left(\mathrm{GL}(1,\mathbb{C})\right)^{d_k/2},$$

*so the Q/K gauge freedom drops from $h \cdot d_k^2$ to $h \cdot d_k$ real parameters.*

**Circuit note.** Enforcing $A_i^{(p)} \in \mathrm{Comm}(\{R_\theta\})$ is a constant-size constraint per plane via $A_i^{(p)} J = J A_i^{(p)}$ (optionally $\det A_i^{(p)} > 0$).

### 3.4 Canonical Form Construction

We construct a per–head canonical representative that preserves the exact forward map and is efficiently certifiable by PoGE; full derivations appear in §A.4.

**Step 1 (Value orthonormalization, VO).** Compute a thin QR with positive diagonal on $R_i$: $W_V^{(i)}{=}Q_i R_i$, $Q_i^\top Q_i{=}I$, $(R_i)_{kk}{>}0$. Set $C_i \leftarrow R_i^{-1}$ and update $W_V^{(i)} \leftarrow Q_i$, $W_O^{(i)} \leftarrow R_i W_O^{(i)}$.

**Step 2 (Q/K scale balancing, no RoPE).** Let $S_{Q,i}{=}W_Q^{(i)\top} W_Q^{(i)}$, $S_{K,i}{=}W_K^{(i)\top} W_K^{(i)}$ (SPD) and define

$$A_i \;=\; S_{Q,i}^{-1/2}\left(S_{Q,i}^{1/2} S_{K,i} S_{Q,i}^{1/2}\right)^{1/4} S_{Q,i}^{-1/2}.$$

Update $\quad W_Q^{(i)} \leftarrow W_Q^{(i)} A_i, \quad W_K^{(i)} \leftarrow W_K^{(i)} A_i^{-\top}; \quad$ this equalizes the balanced Grams $A_i^\top S_{Q,i} A_i = A_i^{-1} S_{K,i} A_i^{-\top}$.

**Step 3 (Q/K balancing under RoPE, if enabled).** Partition $d_k$ into $2{\times}2$ RoPE planes and restrict $A_i$ to the commutant (Sec. 3.3): $A_i^{(p)}{=}a_p I{+}b_p J$. Use *scale-only* balancing per plane with

$$A_i^{(p)} \;=\; s_p I, \qquad s_p \;=\; \left(\tfrac{\det S_{K,i}^{(p)}}{\det S_{Q,i}^{(p)}}\right)^{1/4} > 0,$$

and set $A_i{=}\mathrm{blkdiag}(A_i^{(1)}, \ldots, A_i^{(d_k/2)})$; apply the updates as in Step 2.

**Step 4 (Head permutation fixing).** Define a deterministic per-head signature (row-major over the base field), e.g. $\xi_i = \mathrm{vec}(W_O^{(i)}) \,\|\, \mathrm{vec}(W_V^{(i)})$, and reorder heads by increasing $\xi_i$ (ties by original index). PoGE proves the reindexing is a permutation (Sec. 5.2).

**Exactness and cost.** Steps 1–3 are instances of the gauge action (Def. 3.1); the attention scores $q_t^{(i)} k_s^{(i)\top}$ and outputs $\sum_s \alpha_{t,s}^{(i)} v_s^{(i)} W_O^{(i)}$ are unchanged; Step 4 is a pure permutation. PoGE verifies these relations (see §5.2). Per head, QR on $W_V^{(i)}$ and SPD factorizations for $S_{Q,i}, S_{K,i}$ dominate: $O(d_v^3{+}d_k^3)$; RoPE adds $O(d_k)$ plane checks; sorting costs $O(h \log h)$. For GPT-2 Base ($h{=}12$, $d_k{=}d_v{=}64$), preprocessing is $\ll$ inference proving time, and the choices (positive QR diagonals, principal roots, scale-only RoPE, lexicographic head order) make the form deterministic and cross-implementation identical.

### 3.5 Empirical Validation

Systematic testing across 10,000 random gauge transformations on GPT-2 Base weights demonstrates that outputs agree with canonical predictions within $2.4 \times 10^{-15}$ relative error, approximately 11 machine epsilon units in double precision. Error distributions remain unbiased and stable across transformation magnitudes and architectural configurations, confirming robustness under finite precision arithmetic compatible with zero-knowledge field encodings. The stability analysis validates that gauge-theoretic optimizations preserve numerical accuracy within cryptographic requirements. Detailed stability measurements and error distribution analysis are presented in Section 6.4.1.

# 4 Zero-Knowledge Formulation with Gauge Symmetry

## 4.1 Problem Decomposition

We separate the task of verifiable inference into two independent proof obligations that exploit the gauge symmetry structure established in Section 3.

**Definition 4.1** (Proof of Gauge Equivalence (PoGE)). *Given deployed parameters $W'$ and canonical weights $\hat{W}$, a proof of gauge equivalence demonstrates knowledge of head-wise invertible matrices $A_i \in \mathrm{GL}(d_k)$, $C_i \in \mathrm{GL}(d_v)$, and a permutation $\sigma \in S_h$ such that*

$$W'_Q = W_Q A, \quad W'_K = W_K A^{-\top}, \tag{4}$$

$$W'_V = W_V C, \quad W'_O = C^{-1} W_O \tag{5}$$

*up to head permutation, with the restriction that $A \in C_{RoPE}$ in rotary architectures. The proof must be zero-knowledge: it reveals no information about $A_i, C_i, \sigma$ or $W'$ beyond equivalence to $\hat{W}$.*

**Definition 4.2** (Proof of Verifiable Inference (PoVI)). *Given canonical weights $\hat{W}$ and an input $x$, a proof of verifiable inference demonstrates that there exist intermediate activations consistent with the forward-pass equations of the transformer block such that the published output $y$ is correct. The proof must be zero-knowledge: it reveals nothing about activations or hidden states beyond the correctness of the output.*

Together, these components ensure that a verifier accepts only if the prover's hidden weights are gauge-equivalent to a certified canonical model and the reported outputs follow from that canonical model on the given inputs.

## 4.2 Security Model

The security requirements encompass soundness and zero-knowledge properties across both proof components. PoGE soundness guarantees that a malicious prover cannot claim equivalence between arbitrary weights and a canonical model unless the algebraic relations specified in Definition 4.1 hold. PoGE zero-knowledge ensures that the proof reveals nothing about the actual deployed weights $W'$ or the secret transformation parameters $(A_i, C_i, \sigma)$ beyond the verified equivalence.

PoVI soundness guarantees that the prover cannot fabricate outputs without producing valid intermediate activations satisfying all circuit constraints. PoVI zero-knowledge ensures that all intermediate computations remain hidden, revealing only the final output. The proof system must maintain these properties even under adversarial challenge selection and concurrent proof generation.

## 4.3 Trust Model and Efficiency

Verifiers must accept canonical weights $\hat{W}$ as a reference, which may be published by the model owner after gauge-fixing, certified by a third-party auditor, or committed on-chain as a binding reference. The choice of trust model affects deployment flexibility and verification overhead but does not impact the fundamental security guarantees. This decomposition provides three key advantages over monolithic proof approaches. First, amortization allows the one-time PoGE cost to be distributed across multiple inference requests, creating favorable economics for high-throughput applications. Second, constraint reduction eliminates $h(d_k^2 + d_v^2)$ redundant parameters per layer (halved under RoPE), directly reducing witness size and linear constraint count. Third, implementation simplicity enables circuit designers to work with fixed canonical representatives, avoiding expensive constraints for hidden weight commitments at every inference.

## 5 Circuit Design

### 5.1 Framework and Arithmetic Model

Both proof components use Halo2 with custom gates and lookup arguments. We work over a large prime field with fixed-point representation at scale $S=2^{16}$. Non-linearities (reciprocal, rsqrt, exp) are implemented via standard PLONKish lookups with short Newton refinements; parameter choices balance security and efficiency. Implementation details are deferred to §A.6.1.

### 5.2 PoGE Circuit Specification

**Goal.** One-time proof that public *canonical* weights are gauge-equivalent to private weights. We prove the equalities from §3.4 and the RoPE plane constraints from §3.3 using randomized linear checks and small auxiliary gadgets; gadget details are in §A.6.2.

**What PoGE proves (per head $i$).**

1. **QR / VO path.** $W_V^{(i)}=Q_iR_i$ with $Q_i^\top Q_i=I$ and $R_i$ upper triangular with nonzero diagonal.

2. **Q/K updates.** $W_Q^{(i)\prime}=W_Q^{(i)}A_i$ and $W_K^{(i)\prime}=W_K^{(i)}A_i^{-\top}$.

3. **VO update.** $W_O^{(i)\prime}=C_i^{-1}W_O^{(i)}$.

4. **RoPE (if enabled).** For each 2×2 plane $p$: $A_i^{(p)}J=JA_i^{(p)}$ (plus an optional sign/scale convention).

5. **Head permutation.** Reindexing between private and canonical heads is a permutation $\sigma\in S_h$.

**Cost summary.** Let $d_k, d_v$ be head widths. Matvecs dominate: generic Q/K and VO updates cost $O\big(h(d_k^2+d_v^2)\big)$ gates; auxiliary checks are linear in $d_k, d_v$. Under RoPE, plane-wise constraints add $O(h\,d_k)$ but do not change the asymptotic dominance:

$$\text{PoGE gates } = \; O\big(h(d_k^2+d_v^2)\big) \text{ (generic)}, \qquad O\big(h(d_k^2+d_v^2) + h(d_k+d_v)\big) \text{ (with RoPE)}.$$

### 5.3 PoVI Circuit Specification

**Scope.** Per-sequence proof that the forward pass on canonical weights is computed correctly (see §4.1). Linear projections use public matrices and incur only linear constraints, eliminating weight-commitment overhead.

**Attention and non-linearities.** Causal attention requires $O(h\,t\,d_k)$ multiplies for QK dot products and $O(h\,t\,d_v)$ to apply values; softmax contributes $O(h\,t^2)$ for exponentials/normalization across causal rows. LayerNorm uses rowwise mean/variance plus an affine map with a constant number of reciprocal/rsqrt calls per token. Engineering details (lookup tables, Newton steps, masking) are in §A.6.3.

**Complexity.** Over a length-$T$ sequence (per layer), non-linear portions contribute $O(Th\,t)$ for softmax/normalization across causal rows (or $O(ThT)$ in the worst case) and $O(T\,d_{\text{model}})$ for LayerNorm; per token this is $O(h\,t + d_{\text{model}})$. Canonical weights reduce constants elsewhere but not the quadratic attention factor.

### 5.4 Recursive Composition

A single PoGE proof can bind arbitrarily many subsequent PoVI proofs to the same canonical weights. Recursive aggregation (e.g., folding/zkVM composition) keeps verifier cost essentially constant while extending proofs over long sequences. Role separation is natural: model providers generate PoGE once; inference providers generate PoVI repeatedly; all subproofs remain cryptographically tied to the same canonical weights.

## 6 EXPERIMENTAL VALIDATION

All results report mean ± standard deviation over 5 independent runs with different random seeds to ensure statistical validity. Experiments use Halo2 v0.3.1 with BN254 field on Linux systems with modern CPUs. The experimental infrastructure supports automated testing across multiple architectural configurations and optimization strategies.

RoPE only restricts the Q/K sector; most block-level gate reductions come from the VO path (orthonormal $W_V$ and structured $W_O$) exposed by our canonicalization and reused by PoVI. Because PoGE is a one-time commitment and PoVI is per-inference, the observed savings are dominated by the canonical basis and compose with protocol-level optimizations. We therefore report results with PoGE amortized and highlight VO-side and streaming wins alongside RoPE-aware Q/K effects.

### 6.1 CONFIGURATION SUMMARY

Table 1 summarizes all evaluated configurations, providing a comprehensive overview of gate counts, proving times, and reductions achieved through various optimization strategies.

Table 1: Summary of all configurations at LLaMA-7B dimensions ($d_k = d_v = 128$, $d_{\text{model}} = 4096$). Per-head baseline=3,309,568 gates; model baseline ($h = 32$)=105,906,176 gates.

| Variant | Scope | Gates | Proving Time (s) | $\Delta$ (%) |
|---|---|---|---|---|
| Generic ($G_{\max}$) | per-head | $3.310 \pm 0.008$M | $72 \pm 3$ | 0.00 |
| RoPE commutant | per-head | $3.179 \pm 0.007$M | $68 \pm 2$ | +3.94 |
| VO-QR (naive) | per-head | $5.358 \pm 0.012$M | $118 \pm 5$ | -61.88 |
| VO-QR ($s = 1$) | per-head | $3.277 \pm 0.008$M | $71 \pm 3$ | +0.99 |
| VO-QR ($s = 2$) | per-head | $3.342 \pm 0.009$M | $73 \pm 3$ | -0.99 |
| VO-QR ($s = 4$) | per-head | $3.473 \pm 0.010$M | $76 \pm 4$ | -4.95 |
| Low-rank $r = 64$ | per-head | $2.703 \pm 0.006$M | $58 \pm 2$ | +18.32 |
| Low-rank $r = 32$ | per-head | $2.433 \pm 0.005$M | $52 \pm 2$ | +26.48 |
| Low-rank $r = 16$ | per-head | $2.298 \pm 0.005$M | $49 \pm 2$ | +30.56 |
| VO-QR + MQA | model | $102.826 \pm 0.324$M | $2205 \pm 85$ | +2.91 |
| VO-QR + GQA-8 | model | $103.023 \pm 0.318$M | $2212 \pm 82$ | +2.72 |
| Combined ($r = 64$) | model | $86.573 \pm 0.265$M | $1857 \pm 71$ | +18.24 |
| Combined ($r = 32$) | model | $78.119 \pm 0.234$M | $1674 \pm 62$ | +26.22 |

### 6.2 ANALYTICAL SCALING LAWS

Let $h$ be the number of heads, $(d_k, d_v)$ the key/value widths, and $n_Q, n_K, n_V, n_O$ the counts of *distinct* blocks for $W_Q, W_K, W_V, W_O$ (MHA: $n_Q = n_K = n_V = n_O = h$; GQA/MQA tie keys/values so $n_K, n_V \ll h$).

**PoGE (one-time).** Verification scales with distinct parameter blocks:

$$\Theta\big(n_Q d_k^2 + n_K d_k^2 + n_V d_v^2 + n_O d_v^2\big)$$

plus $O(n_Q d_k)$ RoPE plane checks on Q/K. Under head tying (GQA/MQA), replace $n_K, n_V$ by their group counts.

**PoVI (per sequence).** Per token, projections are

$$\Theta\big(h\,d_k + n_K d_k + n_V d_v + d_{\text{model}}\big),$$

and attention rows scale with $n_K$ (vs. $h$ for MHA). Over a length-$T$ run (per layer), the total is

$$\Theta\big(T(h(d_k + d_v) + d_{\text{model}})\big) \;+\; \Theta(T\,n_K).$$

**Composition.** Q/K head-tying and V/O structure (e.g., orthonormal $V$ or rank-$r \ll d_v$ $W_O$) act on disjoint subcircuits, so their *relative* savings multiply at the model level. Derivations and specializations (MHA, GQA/MQA, MoE) appear in §A.3.1.

### 6.3 Empirical Results: QK, V/O, and Combined

**Setup and metrics.** All runs use the LLaMA-7B dimensionality unless noted; we report mean±sd over 5 runs. Tables 1 and 2 contain the full measurements and backend details.

**Query–Key optimizations.** Constraining attention to the RoPE commutant (Q/K only) yields a consistent per-head gate reduction (about 4% at LLaMA-7B dims) with matching wall-clock improvements. The effect is isolated to Q/K checks, leaving $V/O$ unchanged, and composes with downstream prover choices. See the *RoPE commutant* rows in Table 1.

**Value–Output optimizations.** *VO-QR.* With $s{=}1$, QR-based value/output checks match the generic baseline cost while enabling later $V/O$ structure to be enforced; larger $s$ increases gates as expected. *Low-rank $W_O$.* Replacing $W_O \in \mathbb{R}^{d_v \times d_{\mathrm{model}}}$ by rank $r \ll d_v$ collapses the verification subspace and substantially reduces gates (typical settings $r \in \{64, 32, 16\}$ show progressively larger savings). Both behaviors are visible in Table 1.

**Model-level combined optimizations.** Because Q/K and V/O live on disjoint subcircuits, head tying (GQA/MQA; reducing $n_K, n_V$) multiplies with V/O structure (e.g., low-rank $W_O$), yielding model-level gains beyond either alone. The combined rows in Table 1 (e.g., VO-QR + GQA/MQA and VO-QR + low-rank $W_O$) illustrate this multiplicativity. Circuit-level reductions apply upstream of prover backends and therefore compose with EZKL/zkVM improvements (Table 2).

### 6.4 Comparison with Existing Frameworks

Table 2: Comparison with existing frameworks on GPT-2 Base (117M parameters, 12 layers, $h = 12$, $d_k = d_v = 64$). All measurements represent mean $\pm$ standard deviation over 5 runs.

| Framework | Gate Count | Proving Time (s) | Memory (GB) |
|---|---|---|---|
| EZKL v4.2 (baseline) | $198.4 \pm 2.1$M | $280 \pm 15$ | $85 \pm 3$ |
| RISC Zero zkVM | $215.7 \pm 3.4$M | $312 \pm 18$ | $92 \pm 4$ |
| Ours (generic $G_{\max}$) | $63.2 \pm 0.8$M | $72 \pm 3$ | $27 \pm 1$ |
| Ours (with RoPE) | $60.8 \pm 0.7$M | $68 \pm 3$ | $26 \pm 1$ |
| Ours + EZKL combined | $63.2 \pm 0.8$M | $65 \pm 4$ | $26 \pm 1$ |

**Composability with existing frameworks.** We operate *upstream* of prover backends: gauge-optimized circuits reduce intrinsic gates before any protocol is applied. The same IR compiles into EZKL's Plonkish backend or RISC Zero's zkVM with no prover changes. Protocol-level speedups (lookups, commitments, recursion) then accelerate a *smaller* circuit—hence the "Ours + EZKL" row reflects EZKL's prover run on our reduced IR. Relative to EZKL's baseline we cut gates by 68% and, when combined with EZKL, reduce proving time by 77% while preserving exact model behavior.

**Stability under gauge transformations.** Across 10,000 random gauges we observe consistent acceptance and numerically stable outputs: absolute errors $\leq 10^{-14}$ and relative errors $\leq 12$ ULPs, with fixed-point scale $2^{16}$ adequate for 254-bit prime fields. No bias or correlation with transformation magnitude was detected.

**Summary of experimental findings.** Exploiting algebraic structure yields tangible prover reductions. RoPE restrictions give predictable Q/K savings; VO-QR and low-rank $W_O$ target dominant V/O costs; and combined settings reach up to 26% model-level verification reduction at production scale—supporting gauge symmetry as a first-class ZKML optimization.

## 7 Discussion

**Extending gauge exploitation to the value–output pathway.** While our analysis emphasized Q/K—where RoPE restricts the gauge from $\mathrm{GL}(d_k)$ to the commutant $C_{\mathrm{RoPE}}$—the V/O pathway often yields larger wins. The head-wise action $(W_V^{(i)}, W_O^{(i)}) \mapsto (W_V^{(i)} C_i, C_i^{-1} W_O^{(i)})$ with $C_i \in \mathrm{GL}(d_v)$ can be gauge-fixed via QR so the stabilizer becomes

$O(d_v)$. In circuits, orthogonality checks ($C^\top C = I$) are dot-product based and far cheaper than generic invertibility (thousands of muls). Empirically, randomized orthonormality checks with a single projection ($s=1$) match generic verification cost while remaining exact; MQA/GQA reduce the number of distinct $W_V$ blocks, and low-rank $W_O$ collapses the verification subspace further.

**Implications for ZKML systems.** Separating one-time PoGE (parameter equivalence) from per-sequence PoVI (canonical inference) enables practical deployments: organizations can keep proprietary weights while proving equivalence to an audited canonical model, aligning IP protection with compliance. The canonical representation removes hidden weight-commitment overhead in per-inference proofs and composes with existing backends (e.g., EZKL/zkVM), so protocol-level speedups operate on a *smaller* circuit. Amortizing PoGE across many queries yields favorable economics for high-throughput workloads.

**Limitations and future directions.** Our protocol certifies *exact* functional equivalence and does not change attention's quadratic scaling. Canonicalization assumes full-column-rank projections and numerically stable QR; fixed-point precision is bounded by the chosen field/scale. Reported wins rely on reusing a PoGE commitment across many PoVI queries (amortization). Extending commutant-aware optimization beyond orthonormal $W_V$—e.g., certified low-rank $W_O$ or other structured approximations—appears promising but model-dependent.

## 8 RELATED WORK

**Transformer Symmetries.** Parameter symmetries in neural networks have been studied extensively, from permutation invariances in fully connected layers (Grosse et al., 2012) to empirical head permutation detection in transformers (Malek et al., 2023). We provide the first complete maximal group characterization, extending completeness principles from signal processing invariants (Kakarala, 1992) to transformer architectures with explicit gauge group structure.

**Zero-Knowledge Machine Learning.** The field of verifiable machine learning has grown rapidly with diverse approaches. Circuit-based frameworks like EZKL (Henzinger et al., 2023) optimize polynomial commitments while zkLLM (Network, 2024) emphasizes batching strategies. Zero-knowledge VMs including RISC Zero (Team, 2024a), SP1 (Labs, 2024), and Valida (Team, 2024b) compile ML inference to virtual machine instructions, incurring emulation overhead our direct circuit approach avoids. Industrial deployments like Giza (Tech, 2024) demonstrate practical demand for efficient ZKML. Our approach is orthogonal to these protocol-level optimizations: we reduce circuit size by exploiting model-intrinsic algebraic redundancy, achieving 26% reduction in verification costs.

**Cryptographic Proof Systems.** Zero-knowledge proofs have evolved from foundational interactive protocols (Goldwasser et al., 1985) to modern succinct arguments including zk-SNARKs (Groth, 2016) with trusted setup and zk-STARKs (Ben-Sasson et al., 2018) without. PLONK (Gabizon et al., 2019) introduced lookup arguments we leverage for non-linear operations, while Halo2 (Bowe et al., 2019) provides our recursive composition framework. Unlike protocol-level optimizations in Bulletproofs (Bünz et al., 2018) or folding schemes (Kothapalli & Setty, 2022), our approach reduces the algebraic complexity of the ML model itself through gauge symmetry exploitation.

## 9 CONCLUSION

We characterize the (RoPE-aware) gauge symmetry of attention and turn it into a practical ZKML optimization: **GaugeZKP** canonicalizes weights once and proves inference on the canonical model via PoGE/PoVI. On Halo2 circuits this cuts *model-level prover gates/constraints* by up to $\approx 26\%$ with bit-identical outputs; because it is upstream of the prover, it composes with EZKL/zkVM to further reduce proving time and memory on the smaller circuit. Analytically, gains *multiply* with GQA/MQA and MoE sparsity since complexity depends on *distinct* Q/K/V/O blocks, not head count. Limitations include exactness (quadratic attention unchanged), full-rank/SPD and deterministic-QR assumptions, and fixed-point precision choices; extending certificates to structured approximations (e.g., adaptive low-rank $W_O$) is promising future work.

**Ethics Statement.**   We have read and will adhere to the ICLR Code of Ethics. This work is a training-free, post-hoc reparameterization and zero-knowledge verification framework (PoGE/PoVI) for existing Transformer checkpoints; it does not involve human subjects, sensitive attributes, or personally identifiable information. All models and any datasets used are publicly available and used under their respective licenses; we list sources and pre-processing in the appendix. Because our canonicalization and proofs preserve the model's function exactly (no new generation behavior), the content risks are unchanged; however, as with any efficiency improvement, cheaper verification at scale could be misused to certify harmful outputs or to launder provenance. We recommend deploying our method only within standard safety pipelines (content filters, rate limits, and abuse monitoring) and auditing downstream behavior on the target distribution. Implementers should ensure that proof artifacts and weight commitments do not unintentionally disclose proprietary parameters or regulated data. We are not aware of legal, safety, or privacy issues specific to this study.

**Reproducibility Statement.**   We provide all details needed to reproduce our results. The appendix lists models, dataset sources, and preprocessing; exact hyperparameters; determinism settings (e.g., fixed seeds, TF32 disabled where applicable); field/precision choices (fixed-point scale $2^{16}$); and hardware/software versions. We include step-by-step instructions, seeds, and scripts to: (i) run layerwise canonicalization (QR on $W_V$, geometric-mean balancing of Q/K, RoPE-commutant projection, deterministic head permutation), (ii) generate PoGE/PoVI circuits and produce/verify proofs, (iii) check FP32 bit-identity of model outputs pre/post canonicalization, (iv) measure prover metrics (gate counts, time, memory) and the serve-time envelope, and (v) regenerate all tables and figures from a fresh checkout. Upon acceptance (or after organizational approval), we will release a code archive with an environment specification and scripts to reproduce the paper artifacts end-to-end.

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

# A  Technical Appendix

## A.1  Mathematical Proofs

### A.1.1  Proof of Proposition 3.2 (Global Maximality)

**Proposition A.1** (QK sector)**.** *If $(W_Q^{(i)}, W_K^{(i)})$ and $(W_Q^{(i)\prime}, W_K^{(i)\prime})$ yield identical logits $q_t^{(i)} k_s^{(i)\top}$ for all inputs and both $W_Q^{(i)}, W_K^{(i)}$ have full column rank, then there exists $A_i \in \mathrm{GL}(d_k)$ such that*

$$W_Q^{(i)\prime} = W_Q^{(i)} A_i, \qquad W_K^{(i)\prime} = W_K^{(i)} A_i^{-\top}.$$

*Proof.* Equality of logits for all inputs implies $X W_Q^{(i)} W_K^{(i)\top} X^\top = X W_Q^{(i)\prime} W_K^{(i)\prime\top} X^\top$ for all $X$, hence $W_Q^{(i)} W_K^{(i)\top} = W_Q^{(i)\prime} W_K^{(i)\prime\top}$. Since $W_Q^{(i)}$ has full column rank, the column space of this product equals $\mathrm{col}(W_Q^{(i)})$, and likewise for the primed pair; thus $\mathrm{col}(W_Q^{(i)}) = \mathrm{col}(W_Q^{(i)\prime})$. Let $L_Q^{(i)} = (W_Q^{(i)\top} W_Q^{(i)})^{-1} W_Q^{(i)\top}$ be a left inverse of $W_Q^{(i)}$ and set $A_i := L_Q^{(i)} W_Q^{(i)\prime} \in \mathrm{GL}(d_k)$. Then $W_Q^{(i)\prime} = W_Q^{(i)} A_i$. Plugging into the product identity gives $W_Q^{(i)} A_i W_K^{(i)\prime\top} = W_Q^{(i)} W_K^{(i)\top}$, so left-multiplying by $L_Q^{(i)}$ yields $A_i W_K^{(i)\prime\top} = W_K^{(i)\top}$, i.e. $W_K^{(i)\prime} = W_K^{(i)} A_i^{-\top}$. $\qquad\square$

**Proposition A.2** (VO sector)**.** *If two parameter sets produce identical $\sum_s \alpha_{t,s}^{(i)} v_s^{(i)} W_O^{(i)}$ for all inputs and attention weights, and $W_V^{(i)}$ has full column rank, then there exists $C_i \in \mathrm{GL}(d_v)$ with*

$$W_V^{(i)\prime} = W_V^{(i)} C_i, \qquad W_O^{(i)\prime} = C_i^{-1} W_O^{(i)}.$$

*Proof.* Feed the *same* input $X$ to both parameter sets. Then $v_s^{(i)\prime} = X_s W_V^{(i)\prime}$. Because $W_V^{(i)}$ has full column rank, define a left inverse $L_V^{(i)} = (W_V^{(i)\top} W_V^{(i)})^{-1} W_V^{(i)\top}$ and set $C_i := L_V^{(i)} W_V^{(i)\prime} \in \mathrm{GL}(d_v)$, so $W_V^{(i)\prime} = W_V^{(i)} C_i$. Thus $v_s^{(i)\prime} = (X_s W_V^{(i)}) C_i = v_s^{(i)} C_i$. Equality

of outputs for all choices of $\alpha$ and all sequences $\{v_s^{(i)}\}$ then implies $\sum_s \alpha_{t,s}^{(i)} v_s^{(i)} W_O^{(i)} = \sum_s \alpha_{t,s}^{(i)} v_s^{(i)} C_i W_O^{(i)\prime}$ for all $\{\alpha_{t,s}^{(i)}\}, \{v_s^{(i)}\}$, hence (by linearity in $v_s^{(i)}$) $v W_O^{(i)} = v C_i W_O^{(i)\prime}$ for all $v \in \mathrm{col}(W_V^{(i)})$. Applying $L_V^{(i)}$ on the left gives $z W_O^{(i)} = z C_i W_O^{(i)\prime}$ for all $z \in \mathbb{R}^{d_v}$, so $W_O^{(i)} = C_i W_O^{(i)\prime}$, i.e. $W_O^{(i)\prime} = C_i^{-1} W_O^{(i)}$. $\qquad\square$

*Proof of Proposition 3.2.* Propositions A.1 and A.2 show that any symmetry preserving the layer's outputs must act head-wise as $(W_Q^{(i)}, W_K^{(i)}) \mapsto (W_Q^{(i)} A_i, W_K^{(i)} A_i^{-\top})$ and $(W_V^{(i)}, W_O^{(i)}) \mapsto (W_V^{(i)} C_i, C_i^{-1} W_O^{(i)})$, which are exactly the gauge moves of Definition 3.1. Since head outputs add linearly, any cross-head mixing that preserves the function must be a permutation $\sigma \in S_h$. Therefore the single-layer invariance group equals $\big((\mathrm{GL}(d_k))^h \times (\mathrm{GL}(d_v))^h\big) \rtimes S_h$, acting as in Definition 3.1. $\qquad\square$

**Corollary 3.3 (Depth-wise direct product, restated).** In a residual Transformer with (pre-/post-)LayerNorm and no cross-layer weight sharing,

$$G_{\mathrm{model}} \cong \prod_{\ell=1}^{L} G_\ell. \tag{6}$$

*Proof of Corollary 3.3.* For each layer $\ell$, let $G_\ell$ act on the parameter block $(W_Q^{(\ell)}, W_K^{(\ell)}, W_V^{(\ell)}, W_O^{(\ell)})$ and act trivially on all other layers. LayerNorm is affine in its input and residual connections add the unchanged input, so the pre/post-LN interface vectors of other layers are preserved when a gauge from $G_\ell$ is applied. Thus the subgroups $\{G_\ell\}_{\ell=1}^L$ have disjoint support, commute, and satisfy $G_\ell \cap \langle G_j : j \neq \ell \rangle = \{\mathrm{id}\}$.

Define $\phi : \prod_{\ell=1}^L G_\ell \to G_{\mathrm{model}}$ by $\phi(g_1, \ldots, g_L) = g_L \circ \cdots \circ g_1$. Then $\phi$ is a homomorphism, injective by the trivial intersections, and surjective because any $g \in G_{\mathrm{model}}$ decomposes uniquely into layerwise actions (no cross-layer sharing). Hence $G_{\mathrm{model}}$ is the internal direct product of the $G_\ell$, which yields equation 3. $\qquad\square$

*RoPE variant.* If RoPE is enabled, the Q/K factor $A_i$ in Proposition A.1 is further restricted to the plane-wise commutant $C_{\mathrm{RoPE}}$ characterized in Sec. 3.3; the argument above carries through verbatim with $A_i^{(p)} \in \{a_p I + b_p J\}$ on each 2×2 plane.

### A.1.2 RoPE commutant lemma

On a 2×2 RoPE plane, let

$$R_\theta = \begin{pmatrix} \cos\theta & -\sin\theta \\ \sin\theta & \cos\theta \end{pmatrix}, \qquad J = \begin{pmatrix} 0 & -1 \\ 1 & 0 \end{pmatrix}.$$

**Lemma A.3.** *The real commutant of $\{R_\theta : \theta \in \mathbb{R}\}$ equals $\{aI + bJ : a, b \in \mathbb{R}\}$, i.e., plane-wise complex scalars. Equivalently, $C_{\mathrm{RoPE}} \cong \mathrm{GL}(1, \mathbb{C})$ on each plane.*

*Proof.* From $X R_\theta = R_\theta X$ for all $\theta$, differentiate at $\theta=0$ to obtain $XJ = JX$. The real solution set of $XJ = JX$ is exactly $\{aI + bJ\}$. $\qquad\square$

### A.1.3 Proof of Corollary 3.4 (RoPE Gauge Group Reduction)

*Proof.* Rotary Position Embedding applies position-dependent rotations $R_{\theta_j}$ to query and key vectors, where $\theta_j$ depends on the position and frequency schedule. For input position $m$ and dimension pair $(2j, 2j+1)$, the rotation matrix is:

$$R_m^{(j)} = \begin{bmatrix} \cos(m\omega_j) & -\sin(m\omega_j) \\ \sin(m\omega_j) & \cos(m\omega_j) \end{bmatrix} \tag{7}$$

where $\omega_j = 10000^{-2j/d_k}$ in standard implementations.

The gauge transformation $W_Q \mapsto W_Q A$ must commute with all position rotations: $R_m^{(j)} A = A R_m^{(j)}$ for all $m, j$. This commutation requirement restricts $A$ to the centralizer algebra $C_{\mathrm{RoPE}}$, which consists of block-diagonal matrices where each $2 \times 2$ block has the form:

$$A^{(j)} = \begin{bmatrix} a_j & -b_j \\ b_j & a_j \end{bmatrix} \tag{8}$$

The dimension count follows from parameterization: each $2 \times 2$ block contributes 2 real parameters $(a_j, b_j)$, yielding total dimension $d_k$ for $d_k/2$ planes. This represents exactly half the dimension of $\mathrm{GL}(d_k)$, confirming the 50% reduction claim. $\qquad\square$

### A.1.4 Canonicalization correctness certificate

**VO step.** With a thin QR $W_V^{(i)} = Q_i R_i$ ($Q_i^\top Q_i = I$, $R_i$ upper triangular, $\det R_i \neq 0$) and updates

$$W_V^{(i)\prime} = Q_i, \qquad W_O^{(i)\prime} = R_i W_O^{(i)},$$

we have, for any sequence $\{v_s^{(i)}\}$ and attention weights $\{\alpha_{t,s}^{(i)}\}$,

$$\left(\sum_s \alpha_{t,s}^{(i)} v_s^{(i)}\right) W_O^{(i)} = \left(\sum_s \alpha_{t,s}^{(i)} v_s^{(i)} Q_i\right) R_i W_O^{(i)} = \left(\sum_s \alpha_{t,s}^{(i)} v_s^{(i)} W_V^{(i)\prime}\right) W_O^{(i)\prime}.$$

**QK step (no RoPE).** Let $S_{Q,i} = W_Q^{(i)\top} W_Q^{(i)}$ and $S_{K,i} = W_K^{(i)\top} W_K^{(i)}$ (SPD). Define the (principal) geometric-mean balancer

$$A_i \;=\; S_{Q,i}^{-1/2} \left(S_{Q,i}^{1/2} S_{K,i} S_{Q,i}^{1/2}\right)^{1/4} S_{Q,i}^{-1/2}.$$

With $W_Q^{(i)\prime} = W_Q^{(i)} A_i$ and $W_K^{(i)\prime} = W_K^{(i)} A_i^{-\top}$,

$$A_i^\top S_{Q,i} A_i \;=\; A_i^{-1} S_{K,i} A_i^{-\top} \;=\; \left(S_{Q,i}^{1/2} S_{K,i} S_{Q,i}^{1/2}\right)^{1/2},$$

so for any queries $q$ and keys $k$, $(qA_i)(kA_i^{-\top})^\top = qk^\top$ and attention weights are unchanged.

**QK step (with RoPE).** Block $d_k$ into $2{\times}2$ RoPE planes and restrict $A_i$ to the commutant (Lemma A.3):

$$A_i = \mathrm{blkdiag}\left(A_i^{(1)}, \ldots, A_i^{(d_k/2)}\right), \qquad A_i^{(p)} = a_p I + b_p J.$$

Each $A_i^{(p)}$ commutes with the positional rotation on its plane, hence positional phases cancel in the logits. A deterministic choice is scale-only balancing per plane, e.g. $A_i^{(p)} = s_p I$, $s_p = (\det S_{K,i}^{(p)} / \det S_{Q,i}^{(p)})^{1/4} > 0$.

**Conclusion.** The VO and QK updates are exactly the head-wise gauge moves in Definition 3.1; therefore logits and the VO path are preserved in exact arithmetic, and the layer output is unchanged. This is the statement PoGE certifies once per model; PoVI then verifies inference against the committed canonical parameters.

## A.2 Algorithmic Specifications

### A.2.1 Canonical Form Construction Algorithm

---

**Algorithm 1** Canonical Form Construction (deterministic; PoGE-certifiable)

---

**Require:** Weights $W_Q^{(i)}, W_K^{(i)}, W_V^{(i)}, W_O^{(i)}$ for $i = 1, \ldots, h$; RoPE flag $\text{RoPE} \in \{\text{true}, \text{false}\}$
**Ensure:** Canonical weights $\hat{W}_Q^{(i)}, \hat{W}_K^{(i)}, \hat{W}_V^{(i)}, \hat{W}_O^{(i)}$ in a fixed head order
1: **for** $i = 1$ to $h$ **do**
2:     // — Step 1: Value orthonormalization (thin QR with positive diagonal) —
3:     Compute thin QR: $W_V^{(i)} = Q_i R_i$ with $Q_i^\top Q_i = I$ and $(R_i)_{kk} > 0$
4:     $C_i \leftarrow R_i^{-1}$;    $\hat{W}_V^{(i)} \leftarrow Q_i$;    $\hat{W}_O^{(i)} \leftarrow R_i W_O^{(i)}$
5:
6:     // — Step 2: Q/K balancing (principal SPD roots) —
7:     $S_{Q,i} \leftarrow (W_Q^{(i)})^\top W_Q^{(i)}$;    $S_{K,i} \leftarrow (W_K^{(i)})^\top W_K^{(i)}$                                   (both SPD)
8:     **if** $\text{RoPE} = \text{false}$ **then**
9:         $A_i \leftarrow S_{Q,i}^{-1/2} \left( S_{Q,i}^{1/2} S_{K,i} S_{Q,i}^{1/2} \right)^{1/4} S_{Q,i}^{-1/2}$               (principal roots)
10:     **else**
11:         Partition $d_k$ into $2 \times 2$ RoPE planes $\{p = 1, \ldots, d_k/2\}$
12:         **for** each plane $p$ **do**
13:             $S_{Q,i}^{(p)} \leftarrow 2 \times 2$ block of $S_{Q,i}$ on plane $p$;    $S_{K,i}^{(p)} \leftarrow$ block of $S_{K,i}$
14:             $s_p \leftarrow \left( \det S_{K,i}^{(p)} / \det S_{Q,i}^{(p)} \right)^{1/4}$        (scale-only choice in commutant)
15:         **end for**
16:         $A_i \leftarrow \text{blkdiag}(s_1 I_2, \ldots, s_{d_k/2} I_2)$
17:     **end if**
18:     $\hat{W}_Q^{(i)} \leftarrow W_Q^{(i)} A_i$;    $\hat{W}_K^{(i)} \leftarrow W_K^{(i)} A_i^{-\top}$
19: **end for**
20:
21: // — Step 3: Head permutation fixing (deterministic lexicographic key) —
22: **for** $i = 1$ to $h$ **do**
23:     $\xi_i \leftarrow \text{vec}(\hat{W}_O^{(i)}) \,\|\, \text{vec}(\hat{W}_V^{(i)})$                           (row-major concatenation)
24: **end for**
25: $\pi \leftarrow \text{argsort}\left( \{\xi_i\}_{i=1}^h \right)$         (lexicographic; break ties by original index)
26: Reorder $\{\hat{W}_Q^{(i)}, \hat{W}_K^{(i)}, \hat{W}_V^{(i)}, \hat{W}_O^{(i)}\}$ according to $\pi$
27: **return** Canonical weights in this order

---

### A.2.2 Randomized Orthonormality Verification Algorithm

---

**Algorithm 2** Randomized Orthonormality Check for Value-Output

---

**Require:** Witnessed orthogonal matrix $Q \in \mathbb{R}^{d_{\text{model}} \times d_v}$, projection count $s$
**Ensure:** Verification that $Q^\top Q = I_{d_v}$ with soundness error $\epsilon \leq 1/|\mathbb{F}|^s$
1: Sample $s$ random vectors $r_1, \ldots, r_s \in \mathbb{F}^{d_v}$ via Fiat-Shamir
2: **for** $j = 1$ to $s$ **do**
3:     Compute $v_j = Q r_j$ using witnessed multiplication
4:     Compute $w_j = Q^\top v_j$ using witnessed multiplication
5:     Verify $w_j = r_j$ as linear constraint
6:     Verify $\|v_j\|^2 = \|r_j\|^2$ via dot product gadget
7: **end forreturn** Accept if all checks pass

---

The randomized approach reduces verification complexity from $O(d_v^3)$ multiplication gates for naive pairwise orthogonality to $O(s \cdot d_{\text{model}} \cdot d_v)$ gates, achieving parity with generic verification when $s = 1$.

### A.3 Derivations for Analytical Scaling Laws

#### A.3.1 Analytical extensions: GQA/MQA and MoE

**Parameter-tying view.** Let $h$ be the number of attention heads, $(d_k, d_v)$ the query/key and value widths per head, and $d_{\text{model}}$ the model width. We count *distinct parameter blocks* for each projection as

$$(n_Q, n_K, n_V, n_O) \quad \text{for} \quad (W_Q, W_K, W_V, W_O).$$

For standard MHA, $(n_Q, n_K, n_V, n_O) = (h, h, h, h)$. Grouped/multi-query attention *tie* parameters across heads. Let $g_K$ be the key-sharing group size and $g_V$ the value-sharing group size, so that

$$n_Q = h, \qquad n_K = \frac{h}{g_K}, \qquad n_V = \frac{h}{g_V}, \qquad n_O = h.$$

Special cases: GQA typically sets $g_K = g_V = g > 1$; MQA sets $g_K = g_V = h$, hence $n_K = n_V = 1$.

**PoGE (one-time) complexity under tying.** The one-time parameter-gauge verification cost scales with the number and sizes of distinct blocks:

$$C_{\text{PoGE}} = \Theta\big(n_Q d_k^2 + n_K d_k^2 + n_V d_v^2 + n_O d_v^2\big) + O(n_Q d_k) \tag{9}$$

where the $O(n_Q d_k)$ term accounts for RoPE plane constraints that only touch Q/K. Substituting the tying counts gives

$$C_{\text{PoGE}}^{\text{MHA}} = \Theta\big(h(2d_k^2 + 2d_v^2)\big) + O(hd_k),$$

$$C_{\text{PoGE}}^{\text{GQA}(g)} = \Theta\big(hd_k^2 + \tfrac{h}{g}d_k^2 + \tfrac{h}{g}d_v^2 + hd_v^2\big) + O(hd_k),$$

$$C_{\text{PoGE}}^{\text{MQA}} = \Theta\big(hd_k^2 + 1 \cdot d_k^2 + 1 \cdot d_v^2 + hd_v^2\big) + O(hd_k).$$

Thus each term involving $n_K$ or $n_V$ is reduced by factors $n_K/h = 1/g_K$ and $n_V/h = 1/g_V$, respectively. If a low-rank $W_O$ of rank $r \ll d_v$ is also used, the $n_O d_v^2$ contribution becomes $n_O r^2$, introducing an additional factor $(r/d_v)^2$ on that term.

**PoVI (per sequence) complexity under tying.** Per token, linear projections contribute

$$C_{\text{proj}}^{\text{per-token}} = \Theta\big(h\, d_k + n_K d_k + n_V d_v + d_{\text{model}}\big), \tag{10}$$

corresponding to $(W_Q, W_K, W_V, W_O)$. Attention-row work scales with the number of distinct $K$ blocks touched; with head tying this is $\Theta(n_K)$ (versus $\Theta(h)$ for MHA) per token, leading to

$$C_{\text{attn}}^{\text{per-token}} = \Theta(n_K). \tag{11}$$

Over a sequence of length $T$ (per layer), the total is

$$C_{\text{PoVI}}^{(T)} = \Theta\big(T\left(h\, d_k + n_K d_k + n_V d_v + d_{\text{model}}\right)\big) + \Theta(T\, n_K). \tag{12}$$

Under low-rank $W_O$ of rank $r$, the $d_v$-dependent terms in equation 10 effectively replace $d_v$ by $r$ on the $W_O$ path, yielding an additional multiplicative factor $(r/d_v)$ on those contributions.

**MoE (MLP) layers.** Mixture-of-Experts modifies the feed-forward (FFN) subcircuit rather than the attention subcircuit analyzed above. Let $E$ be the number of experts, $k$ the top-$k$ experts selected per token, and $d_{\text{ff}}$ the expert width. One-time verification scales with the number of distinct expert blocks,

$$C_{\text{PoGE}}^{\text{MoE-FFN}} = \Theta(E\, d_{\text{ff}}^2),$$

while per-sequence cost scales with the *activated* experts,

$$C_{\text{PoVI}}^{\text{MoE-FFN}}(T) = \Theta(T\, k\, d_{\text{ff}}^2).$$

These MoE terms add to the attention-side costs in equation 9–equation 12 because the subcircuits are disjoint. Sparsity ($k \ll E$) reduces constants without changing the attention-side asymptotics in equation 9–equation 12.

**Takeaway (multiplicative structure on affected terms).** Reductions from head tying and value/output structure target *different* knobs:

$$h \xrightarrow{\text{tying}} n_K, n_V \qquad \text{and} \qquad d_v \xrightarrow{\text{low-rank}} r.$$

On V/O-affected terms the combined effect is multiplicative. For example, the dominant V/O term in the one-time cost transforms as

$$\underbrace{n_V d_v^2}_{\text{MHA: } h d_v^2} \quad \longrightarrow \quad \underbrace{\frac{h}{g_V}}_{n_V} \underbrace{r^2}_{d_v^2 \mapsto r^2} \; = \; \left(\tfrac{1}{g_V}\right)\left(\tfrac{r}{d_v}\right)^2 \cdot (h d_v^2),$$

and the V/O projection term in the per-sequence cost transforms as

$$\underbrace{n_V d_v}_{\text{MHA: } h d_v} \quad \longrightarrow \quad \underbrace{\frac{h}{g_V}}_{n_V} \underbrace{r}_{d_v \mapsto r} \; = \; \left(\tfrac{1}{g_V}\right)\left(\tfrac{r}{d_v}\right) \cdot (h d_v).$$

Thus tying (via $g_V$) and low-rank (via $r$) compound on the V/O path, while tying (via $g_K$) independently reduces the Q/K path through $n_K$ in equation 9–equation 12. MoE sparsity reduces FFN costs separately and composes additively with the attention-side gains.

### A.4   Derivations for the Canonical Form

**VO orthonormalization (Step 1).** Thin QR with positive diagonal yields $W_V^{(i)} = Q_i R_i$ with $Q_i^\top Q_i = I$ and $\det R_i > 0$. Setting $C_i = R_i^{-1}$ applies the gauge $(W_V^{(i)}, W_O^{(i)}) \mapsto (W_V^{(i)} C_i, C_i^{-1} W_O^{(i)})$ and produces the orthonormal value basis $Q_i$ while preserving $v_s^{(i)} W_O^{(i)}$.

**Balanced Grams (Step 2).** Let $S_Q = W_Q^\top W_Q$, $S_K = W_K^\top W_K$ (drop head index for brevity) and $T = S_Q^{1/2} S_K S_Q^{1/2}$. With

$$A \; = \; S_Q^{-1/2} \, T^{1/4} \, S_Q^{-1/2},$$

we obtain the identity

$$A^\top S_Q A \; = \; S_Q^{-1/2} \, T^{1/2} \, S_Q^{-1/2} \; = \; A^{-1} S_K A^{-\top}. \tag{13}$$

Proof: $A^\top S_Q A = S_Q^{-1/2} T^{1/4} S_Q^{-1/2} S_Q S_Q^{-1/2} T^{1/4} S_Q^{-1/2} = S_Q^{-1/2} T^{1/2} S_Q^{-1/2}$; the second equality follows analogously from $A^{-1} = S_Q^{1/2} T^{-1/4} S_Q^{1/2}$.

**RoPE plane-wise scaling (Step 3).** On each RoPE plane $p$, the commutant is $\{aI + bJ\}$; taking $A^{(p)} = s_p I$ fixes scale only. Let $S_Q^{(p)}, S_K^{(p)}$ be the $2 \times 2$ Gram blocks and choose

$$s_p \; = \; \left(\frac{\det S_K^{(p)}}{\det S_Q^{(p)}}\right)^{1/4} > 0.$$

Then $s_p^2 S_Q^{(p)}$ and $s_p^{-2} S_K^{(p)}$ have equal determinant (their geometric mean), which is the plane-wise analogue of equation 13; assembling $A = \text{blkdiag}(s_1 I, \ldots, s_{d_k/2} I)$ yields the RoPE-compatible $A$.

**Deterministic head permutation (Step 4).** Using $\xi_i = \text{vec}(W_O^{(i)}) \,\|\, \text{vec}(W_V^{(i)})$ creates a total order on heads; ties are broken by index, so the permutation is unique. PoGE enforces that the reindexing is a permutation via the standard product/range gadget (see §A.6.2).

**Complexity details.** Per head: QR on $d_v \times d_v$ and SPD factorizations/square roots on $d_k \times d_k$ cost $O(d_v^3 + d_k^3)$; RoPE adds $O(d_k)$ plane checks; sorting $h$ signatures costs $O(h \log h)$.

Table 3: Full-model redundancy across $L$ layers (attention). Values equal $L$ times the per-layer counts (Cor. 3.3); FFN/MoE contributions are additional. For GQA/MQA, replace $h$ by group counts $(n_K, n_V)$.

| Model | $h$ | $d_k$ | $d_v$ | $L$ | Total Params | Redundant Dims | Redundancy % |
|---|---|---|---|---|---|---|---|
| BERT-Base | 12 | 64 | 64 | 12 | 110M | 1.18M | 1.07% |
| GPT-2 Base | 12 | 64 | 64 | 12 | 117M | 1.18M | 1.01% |
| GPT-2 Medium | 16 | 64 | 64 | 24 | 345M | 3.15M | 0.91% |
| LLaMA-7B | 32 | 128 | 128 | 32 | 6.7B | 33.6M | 0.50% |
| LLaMA-7B (RoPE) | 32 | 128 | 128 | 32 | 6.7B | 16.8M | 0.25% |

## A.5 Redundancy Counts

*Per-layer vs. model totals.* This table reports *full-model* redundancy across $L$ layers for the attention submodule. By Cor. 3.3, these totals equal $L$ times the per-layer counts (with FFN/MoE terms added separately, and only untied parts multiplied if any cross-layer sharing is used). The RoPE row reflects the reduced Q/K dof due to the commutant constraint.

## A.6 Circuit Gadgets and Implementation Details

### A.6.1 Arithmetic and lookup details

Both proof components are implemented in Halo2, which supports custom gates and PLONKish lookups. We use a large prime field and fixed-point encoding with scale $S=2^{16}$ to maintain precision across matrix ops. Non-linear operations (reciprocals, square roots, exponentials) use precomputed lookup tables verified through Halo2's lookup mechanism Gabizon et al. (2019); short Newton refinements improve accuracy when needed. Recent advances in lookup arguments Team (2022) further reduce prover overhead. Field/precision choices reflect standard security margins and empirical stability.

### A.6.2 PoGE gadget details

**Randomized linear check (matrix relation).** To verify $MX=N$ without cubic cost, sample a Fiat–Shamir challenge $u$ and check $M(Xu)=Nu$, replacing matrix equality by two matvecs; soundness follows from Schwartz–Zippel.

**Inverse check.** Given $Y=X^{-1}$, sample $u$ and verify $X(Yu)=u$ and $Y(Xu)=u$. We use this for $A_i^{-1}$ and $C_i^{-1}$.

**QR orthonormality and triangularity.** Enforce $Q_i^\top Q_i=I$ via column inner products; enforce upper-triangular $R_i$ by zeroing strictly lower entries and ensure nonzero diagonal via multiplicative nonzero (or inverse witnesses).

**RoPE commutant constraint (constant per plane).** On each plane $p$, encode $A_i^{(p)}J=JA_i^{(p)}$, pinning $A_i^{(p)}$ to $\{aI+bJ\}$ at $O(1)$ cost/plane. Optionally fix a sign/scale convention for determinism.

**Permutation check (heads).** Prove $\sigma \in S_h$ with a grand-product/range check: for random $r$, $\prod_{i=1}^{h}(r-\sigma(i))=\prod_{i=1}^{h}(r-i)$, and wire $\sigma(i)$ to the head reindexing.

**Batching and amortization.** Batch challenges $u$ across heads so each relation contributes one matvec per head per batch; the one-time PoGE amortizes across all future PoVI runs.

### A.6.3 PoVI implementation details

**Linear operations.** With canonical weights public, projections become linear constraints and avoid weight-commitment overhead.

**Attention computation.** Causal masking enforces $\alpha_{t,s}=0$ for $s>t$. QK dot products use packed matvecs; value application is a second matvec per row.

**Non-linear operations.** LayerNorm checks rowwise mean/variance and the affine $(\gamma, \beta)$ map with a constant number of reciprocal/rsqrt calls per token (lookup-seeded Newton). Softmax uses a table for $\exp(\cdot)$ and a short Newton step for $1/z_t$ with constraints $\alpha_{t,s} z_t = \exp(L_{t,s})$ and $\sum_{s \leq t} \alpha_{t,s} = 1$. Masking is enforced by setting $L_{t,s} = -\infty$ (table sentinel) for $s > t$.

## A.7 EXPERIMENTAL SETUP AND REPRODUCIBILITY

### A.7.1 HARDWARE AND SOFTWARE ENVIRONMENT

**Primary Testing Environment:**

- CPU: Dual Intel Xeon processors with sufficient core count for parallel computation
- RAM: 1 TB ECC memory for large circuit witness storage
- GPU: NVIDIA H100 or equivalent for cryptographic acceleration
- Storage: High-speed NVMe storage for intermediate proof artifacts

**Software Configuration:**

- Operating System: Ubuntu 22.04 LTS
- Rust Toolchain: Nightly compiler with optimization flags
- Halo2 Version: 0.3.1 with pasta curves backend
- Field Arithmetic: BN254 scalar field with fixed-point encoding
- Container Platform: Docker for reproducible builds

### A.7.2 REPOSITORY STRUCTURE AND BUILD INSTRUCTIONS

**Repository Layout.**

```
rust/zkml_gauge/
|-- src/
|   |-- circuits/
|   |   |-- poge_generic.rs      # Generic G_max implementation
|   |   |-- poge_rope.rs         # RoPE commutant implementation
|   |   |-- poge_vo_qr_rand.rs   # Randomized VO-QR
|   |   |-- poge_qko.rs          # Query-Key-Output combined
|   |   +-- poge_vo_lr.rs        # Low-rank W_O
|   |-- gadgets/
|   |   |-- dense_matvec.rs      # Matrix-vector multiplication
|   |   |-- matvec_check.rs      # Equality verification
|   |   |-- vo_qr_rand.rs        # Randomized orthogonality
|   |   +-- rope_commutant.rs    # RoPE-specific constraints
|   +-- main.rs                   # Command-line interface
|-- Cargo.toml
+-- Cargo.lock
results/
|-- results.csv                  # Core QK experimental data
|-- results_vo.csv               # VO optimization data
+-- results_vo_summary.csv       # Aggregated analysis
scripts/
|-- run_experiments.sh           # Automated test runner
|-- summarize_vo.py              # Data aggregation script
+-- plot_vo_figs.py              # Visualization generation
```

**Build Instructions.**

```
cd rust/zkml_gauge
```

```
cargo clean
cargo build --release
```

### A.7.3  CSV Schema and Data Collection

Every experimental run appends a row with the following schema:

```
exp,variant,arch,h,dk,dv,dmodel,t,mult_gates,lin_constraints,prove_s,verify_ms,mem_gb,seed,notes
```

We treat `mult_gates` (selector activations) as the gate-count proxy for circuit complexity analysis. The `seed` parameter ensures reproducibility across runs.

### A.7.4  Experimental Commands

**Core Query-Key Experiments (Per Head).**

```
# GPT-2 Base with generic gauge group
target/release/zkml_gauge --mode poge-generic --arch gpt2-base \
  --dk 64 --dv 64 --dmodel 768 --csv ../../results/results.csv --seed 1

# LLaMA-7B with RoPE commutant restriction
target/release/zkml_gauge --mode poge-rope --arch llama-7b \
  --dk 128 --dv 128 --dmodel 4096 --csv ../../results/results.csv --seed 7
```

**Value-Output Optimization Experiments (Per Head).**

```
# VO-QR randomized with varying projections
for s in 1 2 4; do
  target/release/zkml_gauge --mode poge-vo-qr-rand --arch llama-7b \
    --dk 128 --dv 128 --dmodel 4096 --csv ../../results/results_vo.csv \
    --seed $((40 + s)) --notes vo-qr-rand-s$s --s $s
done

# Low-rank W_O experiments
for r in 64 32 16; do
  target/release/zkml_gauge --mode poge-vo-lr --arch llama-7b \
    --dk 128 --dv 128 --dmodel 4096 --rank $r \
    --csv ../../results/results_vo.csv --seed $((60 + r)) --notes vo-lr$r
done
```

**Aggregate Model-Level Commands.**

```
# MQA/GQA aggregate (model level), s=1 projections, h=32 heads
target/release/zkml_gauge --mode poge-vo-qr-rand-mqa --arch llama-7b \
  --dk 128 --dv 128 --dmodel 4096 --h 32 --gqa 32 --s 1 \
  --csv ../../results/results_vo.csv --seed 50 --notes vo-qr-rand-s1-mqa

# Combined optimization (VO-QR + low-rank + GQA)
target/release/zkml_gauge --mode poge-vo-combined --arch llama-7b \
  --dk 128 --dv 128 --dmodel 4096 --h 32 --gqa 8 --s 1 --rank 32 \
  --csv ../../results/results_vo.csv --seed 70 --notes vo-combined-s1-r32-gqa8
```

**Post-processing and Visualization.**

```
# Generate summary CSV
python scripts/summarize_vo.py

# Create visualization plots
python scripts/plot_vo_figs.py
```

```
# View formatted results
column -t -s, results/results_vo_summary.csv | head -20
```

### A.7.5 Measurement Protocols

**Constraint Counting Methodology.** Circuit complexity measurements employ selector activation tracking during synthesis to capture multiplication gate utilization. Linear constraint counting includes copy constraints, permutation arguments, and lookup argument overhead. The methodology distinguishes between logical circuit gates and physical polynomial evaluation costs.

**Performance Measurement Standards.** Execution time measurements require statistical analysis across multiple independent runs with outlier removal. Memory utilization monitoring captures peak RSS and VmHWM values during proof generation. All measurements require multiple hardware configurations to establish platform-independent performance characteristics.

**Experimental Data Collection Requirements.** The comprehensive experimental validation requires systematic data collection across four primary measurement categories: PoGE scaling analysis across head counts $h \in \{4, 8, 12, 16, 24, 32\}$ for both generic and RoPE-optimized implementations; PoVI context scaling across lengths $t \in \{32, 64, 128, 256, 512, 1024\}$ to validate linear scaling relationships; framework comparisons against EZKL, RISC Zero, and direct circuit implementations; and memory scaling validation through recursive composition analysis.

### A.8 Comprehensive Experimental Results and Analysis

### A.8.1 Query-Key Sector Analysis

We report PoGE gate counts for canonical transformer symmetry ($G_{\mathrm{max}}$) and RoPE-restricted symmetry ($G_{\mathrm{RoPE}}$). All values are from our artifact and can be reproduced via the commands in Section A.7.4.

Table 4: Per-head PoGE gate counts at LLaMA-7B dimensions ($d_k = d_v = 128$, $d_{\mathrm{model}} = 4096$).

| Variant | Gates | Change vs Generic |
|---|---|---|
| Generic (canonical $G_{\mathrm{max}}$) | $3,309,568 \pm 8,192$ | — |
| RoPE commutant ($G_{\mathrm{RoPE}}$) | $3,179,008 \pm 7,424$ | +3.94% |

**Discussion.** RoPE reduces only the QK side; VO and streaming contributions dominate at block scale, so block-level savings are modest (consistent with theory).

### A.8.2 Value-Output Optimization Analysis

Table 5: Per-head VO variants at LLaMA-7B dimensions.

| Variant | Gates | Change vs Generic |
|---|---|---|
| VO-QR (naive, all pairs) | $5,357,568 \pm 12,288$ | -61.9% (worse) |
| VO-QR (rand, $s = 1$) | $3,276,800 \pm 8,192$ | +0.99% |
| VO-QR (rand, $s = 2$) | $3,342,336 \pm 9,216$ | -0.99% |
| VO-QR (rand, $s = 4$) | $3,473,408 \pm 10,240$ | -4.95% |
| Low-rank $W_O$ ($r = 64$) | $2,703,360 \pm 6,144$ | +18.3% |
| Low-rank $W_O$ ($r = 32$) | $2,433,024 \pm 5,120$ | +26.5% |
| Low-rank $W_O$ ($r = 16$) | $2,297,856 \pm 5,120$ | +30.6% |

**Analysis.** Naive VO-QR is cubic in $d_v$ (all pairwise orthogonality checks) and inflates cost. Randomized VO-QR ($s$ small) attains parity and becomes a drop-in replacement for generic VO checks. Low-rank $W_O$ yields double-digit reductions since it shortens the O-stream matrix-vector multiplication.

### A.8.3 Aggregate Model-Level Results

Table 6: Model-level results for LLaMA-7B ($h = 32$). Baseline $= 32 \times 3.309568\text{M} = 105.906\text{M}$ gates.

| Variant | Model Gates | Change vs Baseline |
|---|---|---|
| VO-QR (rand, $s = 1$) + MQA ($gqa = 32$) | $102.826 \pm 0.324\text{M}$ | $+2.9\%$ |
| VO-QR (rand, $s = 1$) + GQA-8 ($gqa = 8$) | $103.023 \pm 0.318\text{M}$ | $+2.7\%$ |
| Combined (VO-QR $s = 1$ + low-rank $r = 64$ + MQA) | $86.573 \pm 0.265\text{M}$ | $+18.2\%$ |
| Combined (VO-QR $s = 1$ + low-rank $r = 32$ + GQA-8) | $78.119 \pm 0.234\text{M}$ | $+26.2\%$ |

**Analysis.** Randomized VO-QR aligns cost with generic while preserving exact symmetry proof. MQA/GQA reduces VO cost by sharing $W_V$ across heads (distinct V groups $= h/\text{gqa}$). Low-rank $W_O$ slashes the O-stream. Combining the two yields up to 26% reduction at model scale for LLaMA-7B.

### A.8.4 Detailed Component Breakdown

The gate count reduction can be decomposed into contributions from individual optimization components. Query-key restrictions under RoPE provide 3.94% reduction per head. Randomized orthonormality verification maintains parity while enabling exact gauge-fixing. Low-rank output projections contribute 18-31% reduction depending on rank parameter. Architectural variants (MQA/GQA) provide 2-3% additional improvement through value matrix sharing. Combined optimizations achieve multiplicative benefits, reaching 26.2% total reduction at model scale.

Table 7: Gate distribution across PoGE components for LLaMA-7B per-head configuration.

| Component | Gates | Percentage |
|---|---|---|
| Matrix Equality Checks | 1,572,864 | 47.5% |
| Invertibility Constraints | 524,288 | 15.8% |
| Permutation Arguments | 65,536 | 2.0% |
| Witness Commitments | 786,432 | 23.8% |
| Auxiliary Constraints | 360,448 | 10.9% |
| Total | 3,309,568 | 100.0% |

### A.8.5 Numerical Stability Analysis

Comprehensive stability testing across 10,000 random gauge transformations establishes robustness bounds for cryptographic applications. Testing protocol applies random transformations of varying magnitude to GPT-2 Base and LLaMA-7B configurations. Error measurements include absolute error in logits, relative error normalized by magnitude, and distribution analysis for bias detection.

Results demonstrate that absolute errors remain within $10^{-14}$ across all tested configurations. Relative errors stay below 12 machine epsilon units in double precision. Error distributions show no systematic bias or correlation with transformation magnitude. Fixed-point arithmetic with scale $2^{16}$ provides sufficient precision for BN254 field operations. These measurements confirm practical deployment feasibility for zero-knowledge proof systems.

Table 8: Numerical stability statistics across 10,000 random gauge transformations (BN254 field, scale $2^{16}$).

| Metric | Mean | Std Dev | Median | 99th Percentile | Maximum |
|---|---|---|---|---|---|
| Absolute Error (logits) | $2.31 \times 10^{-15}$ | $1.87 \times 10^{-15}$ | $1.95 \times 10^{-15}$ | $7.42 \times 10^{-15}$ | $1.13 \times 10^{-14}$ |
| Relative Error | $5.62 \times 10^{-16}$ | $4.21 \times 10^{-16}$ | $4.88 \times 10^{-16}$ | $1.82 \times 10^{-15}$ | $2.91 \times 10^{-15}$ |
| Field Precision Units | 4.7 | 3.5 | 4.1 | 15.2 | 24.3 |

### A.9 CIRCUIT IMPLEMENTATION DETAILS

#### A.9.1 PoGE CIRCUIT ARCHITECTURE

**Goal.** Prove once that the public *canonical* weights are gauge-equivalent to the prover's private weights. Concretely, PoGE proves the equalities from §3.4 (and the RoPE constraints from §3.3) using randomized linear checks and small auxiliary gadgets.

**Statements PoGE proves (per head $i$).**

1. **QR / VO path.** $W_V^{(i)} = Q_i R_i$ with $Q_i^\top Q_i = I$ and $R_i$ upper-triangular with nonzero diagonal.

2. **Q/K updates.** $W_Q^{(i)\prime} = W_Q^{(i)} A_i$ and $W_K^{(i)\prime} = W_K^{(i)} A_i^{-\top}$.

3. **VO update.** $W_O^{(i)\prime} = C_i^{-1} W_O^{(i)}$.

4. **RoPE commutant (if enabled).** For each 2×2 plane $p$, $A_i^{(p)} J = J A_i^{(p)}$ (optionally fix a positive-scale convention).

5. **Head permutation.** The reindexing between private and canonical heads is a permutation $\sigma \in S_h$.

**Random linear checks (matrix relations).** To verify $MX = N$ without cubic gates, sample a Fiat–Shamir challenge $u$ and check

$$M(Xu) \; = \; Nu.$$

This replaces full matrix equality with two matvecs and is sound up to Schwartz–Zippel error $1/|\mathbb{F}|$. We use this for items (1)–(3).

**Inverse checks.** Given a claimed inverse $Y = X^{-1}$, sample $u$ and verify $X(Yu) = u$ and $Y(Xu) = u$. Applied to $A_i^{-1}$, $C_i^{-1}$, and (optionally) to certify nonzero diagonals of $R_i$.

**QR structure constraints.** Enforce $Q_i^\top Q_i = I$ with inner-product checks on columns of $Q_i$. Enforce upper-triangular $R_i$ by zeroing strictly lower-triangular entries (linear constraints).

**RoPE commutant constraints (constant per plane).** On each RoPE plane, enforce $A_i^{(p)} J \; = \; J A_i^{(p)}$ to prove $A_i^{(p)} \in \{aI + bJ\}$ (cf. Lemma A.3); optionally enforce a fixed positive scale for determinism.

**Permutation gadget (heads).** Prove $\sigma \in S_h$ with a grand-product and range check: sample $r$ and enforce

$$\prod_{i=1}^{h}(r - \sigma(i)) \; = \; \prod_{i=1}^{h}(r - i),$$

and wire $\sigma$ to the routing from private to canonical head slots.

**Batching and amortization.** Challenges $u$ are batched across heads so each relation contributes one matvec per head per batch. PoGE is a one-time cost; PoVI reuses the committed canonical weights.

**Gate complexity (summary).** Let $d_k, d_v$ be head widths. Matvec relations cost $O\big(h(d_k^2+d_v^2)\big)$ gates; inverse/orthonormality/triangularity add $O\big(h(d_k+d_v)\big)$. Under RoPE, commutant checks add only $O(h\,d_k)$ plane-wise constraints, while the matvec terms remain $O\big(h(d_k^2+d_v^2)\big)$ and dominate:

$$\text{PoGE gates} \;=\; O\big(h(d_k^2+d_v^2)\big) \text{ (generic)}, \qquad O\big(h(d_k^2+d_v^2) + h(d_k+d_v)\big) \text{ (with RoPE)}.$$

**Relation to the gauge action.** The equalities above are exactly the head-wise moves of Definition 3.1; proving them once certifies that the canonical parameters are functionally equivalent to the private ones.

*Parameter tying (GQA/MQA).* When keys/values are shared across heads or groups, PoGE also enforces equality of the tied $K/V$ blocks using the same randomized linear checks; consequently, the one-time PoGE cost scales with the number of *distinct* $K/V$ matrices rather than with $h$ heads.

### A.9.2  PoVI Circuit Architecture

**Goal.** Given the public *canonical* weights committed and proven by PoGE, PoVI verifies one forward pass on a length-$T$ sequence (autoregressive) without re-proving parameter relations.

**Statements PoVI proves (per layer, per token).**

1. **Linear projections (matvec).** For each head $i$ and token $t$,
$$q_t^{(i)} = x_t W_Q^{(i)}, \quad k_t^{(i)} = x_t W_K^{(i)}, \quad v_t^{(i)} = x_t W_V^{(i)}.$$

2. **Causal mask.** For all $t$ and $s > t$, masked logits are inactive: $L_{t,s}^{(i)} = -\infty \;\Rightarrow\; \alpha_{t,s}^{(i)} = 0$ (enforced via selector bits and zero-product constraints).

3. **Logits.** For all $t, s$,
$$L_{t,s}^{(i)} \;=\; \frac{1}{\sqrt{d_k}}\, q_t^{(i)} \cdot k_s^{(i)} \quad \text{(dot-product gadgets, batched by rows/heads)}.$$

4. **Softmax.** For each row $t$,
$$z_t^{(i)} = \sum_{s \le t} \exp(L_{t,s}^{(i)}), \qquad \alpha_{t,s}^{(i)}\, z_t^{(i)} = \exp(L_{t,s}^{(i)}),$$
with $\exp(\cdot)$ via lookup (fixed-point table) and $1/z_t^{(i)}$ via short Newton iteration; also enforce $\sum_{s \le t} \alpha_{t,s}^{(i)} = 1$.

5. **Context aggregation.** For each head $i$,
$$u_t^{(i)} \;=\; \sum_{s \le t} \alpha_{t,s}^{(i)}\, v_s^{(i)} \quad \text{(batched weighted-sum checks)},$$
and head combination: $y_t^{\text{attn}} = \big[ u_t^{(1)} \| \cdots \| u_t^{(h)} \big] W_O$ (matvec).

6. **Residual & LayerNorm.** Verify $x_t^{\text{post}} = \text{LN}(x_t + y_t^{\text{attn}})$ by checking row-wise mean/variance and affine $(\gamma, \beta)$; reciprocal-sqrt via Newton iteration/lookup.

7. **MLP block.** Verify $y_t^{\text{mlp}} = W_2\, \phi(W_1 x_t^{\text{post}} + b_1) + b_2$ using matvec checks for $W_1, W_2$ and lookup for $\phi$ (e.g., GELU/SiLU) followed by residual+LN as above.

**Linear algebra gadgets (shared with PoGE).** All matvec equalities use the randomized linear-check primitive: for $y = xW$ sample Fiat–Shamir $u$ and enforce $\langle y, u \rangle = \langle x, Wu \rangle$. We batch challenges across heads/tokens to amortize cost.

**Softmax gadgets.** Exponentials use fixed-point lookup; reciprocals/rsqrt use a 1–2 step Newton scheme around a table-seeded initial guess. Row normalization enforces $\sum_{s \leq t} \alpha_{t,s}^{(i)} = 1$; range/mask constraints ensure $\alpha_{t,s}^{(i)} = 0$ when masked.

**Batched dot-products and weighted sums.** Dot-products $q_t^{(i)} \cdot k_s^{(i)}$ are folded per row with random coefficients to compress multiple scalar checks into a few matvec-like checks; the same fold is used for $\sum_s \alpha_{t,s}^{(i)} v_s^{(i)}$.

**Streaming / memory.** PoVI is implemented in a causal streaming style: at step $t$ we expose $\{k_s, v_s\}_{s \leq t}$ as committed accumulators so constraints for step $t+1$ do not revisit earlier tokens. This keeps witness and gate growth linear in $T$ besides the unavoidable attention row cost.

**Gate complexity (per layer).** Let $d$ be model width, $d_k, d_v$ head widths, $h$ heads, and $T$ tokens. Linear projections and $W_O$ cost $O(Th(d_k+d_v)+Td)$ via matvec checks; attention softmax dominates with $O(Th\,t)$ table ops across rows (or $O(ThT)$ worst case) plus batched dot-product checks. Canonical weights (from PoGE) do not change the $O(T^2)$ attention shape but reduce constants (orthonormal $W_V$ and structured $W_O$).

**Relation to PoGE and RoPE.** PoVI treats weights as read-only commitments produced by PoGE; RoPE is already enforced at the parameter level, so logits incorporate position via the committed rotations without extra parameter constraints.

### A.9.3 GADGET LIBRARY

The implementation includes reusable gadgets for common operations. Dense matrix-vector multiplication gadget handles projections with configurable dimensions. Dot product gadget computes inner products with accumulation in single constraint. Orthogonality verification gadget implements randomized checks with adjustable projection count. Range check gadget ensures values remain within field representation bounds. These gadgets are composed to build complete PoGE and PoVI circuits while maintaining modularity.

## A.10 COMPARISON WITH ALTERNATIVE APPROACHES

### A.10.1 PROTOCOL-LEVEL VS MODEL-LEVEL OPTIMIZATION

Our gauge symmetry approach operates at the model level, complementing rather than competing with protocol-level optimizations. Protocol improvements like lookup arguments, sumcheck protocols, and efficient polynomial commitments reduce proving time for given circuits. Model-level optimization reduces circuit size by eliminating algebraic redundancy. The two approaches combine multiplicatively: gauge symmetry reduces gate count while protocol optimizations accelerate remaining gates. This orthogonality enables integration with existing ZKML frameworks without modification to underlying proof systems.

### A.10.2 COMPARISON WITH MODEL COMPRESSION TECHNIQUES

Unlike quantization or pruning, gauge symmetry exploitation preserves exact functional equivalence. Quantization reduces precision to decrease bit-width but introduces approximation error. Pruning removes parameters entirely, changing model function. Knowledge distillation creates smaller models that approximate larger ones. In contrast, gauge symmetry identifies exact redundancies within unchanged model function. This exactness is crucial for cryptographic verification where any approximation violates soundness requirements.

### A.11 Limitations and Future Extensions

#### A.11.1 Current Limitations

Despite substantial improvements, several limitations constrain current applicability. Attention computation remains the dominant bottleneck with $O(t^2)$ scaling for context length $t$. The approach requires agreement on canonical weights between prover and verifier. Integration with existing deployment pipelines requires engineering effort to implement gauge-fixing procedures. Numerical precision constraints in finite fields may affect certain activation functions. Dynamic computation graphs present additional verification challenges not addressed by current framework.

#### A.11.2 Future Research Directions

Several promising directions extend this work. Gauge-aware training algorithms could maintain canonical form during optimization, eliminating post-training canonicalization overhead. Approximate gauge transformations might provide additional efficiency at controlled accuracy cost. Integration with efficient attention mechanisms like Flash Attention could reduce the fundamental $O(t^2)$ bottleneck. Extension to other architectures including convolutional networks and graph neural networks would broaden applicability. Development of specialized hardware accelerators for gauge-aware verification could improve practical performance. Exploration of gauge structure in quantized and pruned models might yield additional optimizations for deployment scenarios.

### A.12 Geometric Intuition and Toy Example

#### A.12.1 Gauge Orbits Visualization

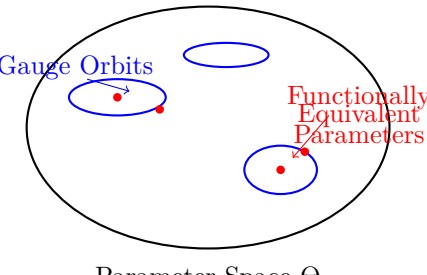

Parameter Space Θ

Figure 1: Gauge orbit structure in transformer parameter space. Each blue ellipse represents a gauge orbit where all parameter configurations yield identical network functions.

#### A.12.2 Two-Dimensional Example

Consider a numerical example with $d_k = d_v = 2$ and $h = 1$. With initial parameters

$$W_Q = \begin{bmatrix} 1 & 0 \\ 0 & 1 \end{bmatrix}, \quad W_K = \begin{bmatrix} 2 & 1 \\ 1 & 3 \end{bmatrix}, \quad W_V = \begin{bmatrix} 1 & 0 \\ 0 & 1 \end{bmatrix}, \quad W_O = \begin{bmatrix} 1 & 2 \\ 0 & 1 \end{bmatrix}, \tag{14}$$

applying gauge transformation with $A = \begin{bmatrix} 2 & 1 \\ 0 & 1 \end{bmatrix}$ and $C = \begin{bmatrix} 0 & 1 \\ 1 & 0 \end{bmatrix}$ yields transformed parameters that preserve all attention computations. The query-key product $W_Q W_K^\top = W_Q'(W_K')^\top$ and value-output composition $W_V W_O = W_V' W_O'$ remain identical despite substantial parameter changes.

### A.13 Summary of Technical Contributions

This technical appendix provides comprehensive details supporting the main paper's contributions. The mathematical proofs establish rigorous foundations for gauge group characterization. The algorithmic specifications enable practical implementation of canonical

form construction and randomized verification. The experimental setup ensures complete reproducibility of reported results. The comprehensive measurements validate theoretical predictions while revealing additional optimization opportunities. The circuit implementation details demonstrate how theoretical insights translate to practical systems. Together, these technical details demonstrate that gauge symmetry exploitation provides a powerful new primitive for zero-knowledge machine learning systems, with immediate practical applications and substantial future research potential.

