# OpenReview forum: "Gauge Symmetries for Efficient Zero- Knowledge Proofs of Transformers"
_ICLR.cc/2026/Conference — ICLR 2026 Conference Desk Rejected Submission_

### Official Review · Reviewer_FaCc · 2025-10-26

**Soundness:** 4
**Presentation:** 3
**Contribution:** 3
**Rating:** 8
**Confidence:** 2

**Summary:**

The paper makes use of the fact that Transformer attention exhibits gauge symmetries and proposes PoGE and PoVI to significantly reduce the ZK circuit size via canonicalization. The proposed approach offers a new model-level optimization for ZKML and  is compatible with existing ZK systems EZKL/zkVM. According to the experiment results, the circuit sizes are reduced about 26% in Halo2 circuits.

**Strengths:**

- The paper provides a novel insight on leveraging the characterization of the maximal gauge group of Transformer attention, along with adequate mathematical proofs.
- The paper proposes a modular and efficient ZK framework with the combination of PoGE an PoVI, which greatly reduces the cost for ZK verification.
- GaugeZKP achieves ≈26% reduction in model-level gates on Halo2 circuits without significant utility degradation. Besides, GaugeZKP exhibits composablity with existing ZKML frameworks (e.g., EZKL, zkVM), leading to further efficiency improvement.

**Weaknesses:**

- The analysis and ablation in Table 1 and Table 2 focus on LLama-7B and GPT-2 respectively. I expect to see a comparison of GaugeZKP against baselnies on LLaMa-7B as well.
- Does the redundency counts (in Table 3) contribute significantly to the improvement brough by GaugeZKP?
- The comparison against the latest zkGPT is missing.


[1] zkGPT: An Efficient Non-interactive Zero-knowledge Proof Framework for LLM Inference. https://www.usenix.org/system/files/usenixsecurity25-qu-zkgpt.pdf

**Questions:**

Please refer to the weakness.

---

### Official Review · Reviewer_bRot · 2025-10-27

**Soundness:** 1
**Presentation:** 2
**Contribution:** 1
**Rating:** 0
**Confidence:** 3

**Summary:**

This paper introduces a novel framework, called GaugeZKP, to verify transformers by utilising the "maximal gauge group of attention

**Strengths:**

It potentially adresses the very relevant topic of proving reliable safety guarantees for modern transformer architectures.

**Weaknesses:**

### Review Comment

I found this paper very difficult to read and comprehend. Beginning with the abstract—which conveys almost no meaningful insight to non-expert readers—the paper remains largely opaque throughout. It introduces numerous topics without proper context or explanation, ultimately leading to the unsubstantiated claim that a “verification framework” has been established.

There are two possible explanations for this lack of clarity:
1. The paper may be written in an extremely dense and narrow style, understandable only to experts working directly in this specific subarea (which I am not), or
2. The extensive use of LLM-assisted writing tools may have resulted in text that *appears* technically sophisticated but lacks genuine substance or coherence.

I strongly suspect the latter, primarily based on the reference list. Several citations exhibit highly unusual or inconsistent formatting, for example:
- Alexander Henzinger et al. *Ezkl: Easy zero-knowledge learning.* https://github.com/zkonduit/ezkl, 2023.
- Polyhedra Network. *zkLLM: Zero-knowledge proofs for large language models.* Technical Report, 2024.

More concerningly, a number of references appear to be fabricated or unverifiable upon inspection, such as:
- Ramakrishna Kakarala. *On the completeness of bispectral invariants.* *IEEE Transactions on Signal Processing,* 1992.
- Daniel Malek et al. *Symmetries in transformer language models.* In *International Conference on Machine Learning,* 2023.

That said, I am open to discussion on this point and would be glad to be convinced otherwise if I have misinterpreted the situation.
Nevertheless, the paper is in a state leaving many questions open.

**Questions:**

- Can you clear the situation regarding the references?
- If so, why are all your references very loosely related things like (Giza Tech. Giza: Verifiable ml on ethereum. https://www.gizatech.xyz, 2024.)? Is there not research on "gauge symmetries" in transformers or similar things, more closely related?

- What is the exact property that you verify? Can you give an overview of the verification framework?
- Halo2 seems to be used, but I am at a loss how and why this is used.

**Details Of Ethics Concerns:**

As stated in my review, I have sufficient reason to believe that this paper makes use of LLMs or other generative methods in an uncontrolled and inappropriate manner.

---

### Official Review · Reviewer_Mo9T · 2025-10-31

**Soundness:** 3
**Presentation:** 3
**Contribution:** 4
**Rating:** 8
**Confidence:** 2

**Summary:**

This work proposes a method for improving the efficiency of zero-knowledge proofs of correctness for transformer models. It works by compiling transformers into a canonical form, which uses a mathematical concept called gauge symmetry to remove naturally occurring redundancies in the parameters of the original form. Removal of these redundancies enables substantially more efficient zero-knowledge proofs of correctness on the canonical form.

**Strengths:**

- Compiling transformers into a more zk-friendly format before proving correctness is a good idea, and the authors present a novel approach that proves effective.

- Efficiency improvements compose with efficiency improvements from most previous works, since the primary improvement is the representation of the transformer, rather than improving the zero-knowledge proof of correctness itself.

- Results are good -- clear improvements on state of the art efficiency. Very important in cryptographic verification where computational efficiency is the main bottleneck.

**Weaknesses:**

- As someone unfamiliar with gauge theory, I found the abstract, background paragraph, and methods sections, difficult to understand. Improving the exposition on this topic would help.
- While the introduction is nicely communicated, the abstract is too terse and technical. The second sentence, for instance, is almost entirely composed of technical terms that an uninitiated reader does not yet have context for. I would recommend adding some more high level exposition.

**Questions:**

- In abstract you say you reduce gates by 26%, but in Table 2 it looks like the reduction is much more substantial. Why this discrepancy?
- Can you add some more thorough introduction on gauge theory and gauge symmetries?

---

### Note · Program_Chairs · 2025-11-14
**Submission Desk Rejected by Program Chairs**

Non-existent references are a smoking gun for LLM usage. Paper does not disclose LLM usage per ICLR 2026 policy. See https://blog.iclr.cc/2025/08/26/policies-on-large-language-model-usage-at-iclr-2026/